# Effectiveness of a Very Low Calorie Ketogenic Diet on Testicular Function in Overweight/Obese Men

**DOI:** 10.3390/nu12102967

**Published:** 2020-09-28

**Authors:** Laura M. Mongioì, Laura Cimino, Rosita A. Condorelli, Maria Cristina Magagnini, Federica Barbagallo, Rossella Cannarella, Sandro La Vignera, Aldo E. Calogero

**Affiliations:** Department of Clinical and Experimental Medicine, University of Catania, Via S. Sofia 78, 95123 Catania, Italy; lauramongioi@hotmail.it (L.M.M.); lauracimino@hotmail.it (L.C.); crymaga@hotmail.it (M.C.M.); federica.barbagallo11@gmail.com (F.B.); rossella.cannarella@phd.unict.it (R.C.); sandrolavignera@unict.it (S.L.V.); acaloger@unict.it (A.E.C.)

**Keywords:** very-low-calorie ketogenic diet, glucose homeostasis, lipid profile, vitamin D, testosterone

## Abstract

Background: Obesity has become an increasingly worrisome reality. A very-low-calorie ketogenic diet (VLCKD) represents a promising option by which to achieve significant weight loss. This study sought to evaluate the effectiveness of VLCKD on metabolic parameters and hormonal profiles of obese male patients. Methods: We enrolled 40 overweight/obese men who consumed VLCKD for at least eight weeks. Body weight, waist circumference, fasting glucose, insulin, total cholesterol, high-density lipoprotein, triglycerides, creatinine, uric acid, aspartate aminotransferase, alanine aminotransferase, vitamin D, luteinizing hormone (LH), total testosterone (TT), and prostate-specific antigen (PSA) were calculated before and after VLCKD consumption. We additionally determined the homeostasis model assessment index and low-density lipoprotein (LDL) values. Results: After VLCKD (13.5 ± 0.83 weeks), the mean body weight loss was 21.05 ± 1.44 kg; the glucose homeostasis and lipid profile were improved significantly; serum vitamin D, LH, and TT levels were increased and the PSA levels were decreased significantly as compared with pretreatment values. These results are of interest since obesity can lead to hypogonadism and in turn, testosterone deficiency is associated with impaired glucose homeostasis, metabolic syndrome, and diabetes mellitus. Moreover, a close relationship between obesity, insulin resistance, and/or hyperinsulinemia and increased prostate volume has been reported, with a consequent greater risk of developing lower urinary tract symptoms. Conclusions: VLCKD is an effective tool against obesity and could be a noninvasive, rapid, and valid means to treat obese patients with metabolic hypogonadism and lower urinary tract symptoms.

## 1. Introduction

In the last two decades, obesity has become an increasingly worrisome reality, with evidence of a dramatic increase in its prevalence, estimated to be around 20% [1]. According to the World Health Organization (WHO), about 1.6 billion adults worldwide were overweight in 2005 and around 400 million were obese [2]. In the United States, it is estimated that about two out of three women and three out of four men are overweight or obese, with the treatment of related diseases costing an average of more than 200 billion dollars annually [3]. Among countries in the European Union, overweightness affects 30% to 70% of the population, while about 10% to 30% of adults are affected by obesity (http://www.euro.who.int/en/health-topics/noncommunicable-diseases/obesity/data-and-statistics). Obesity is often associated with comorbidities such as essential hypertension, type 2 diabetes mellitus (DM2), cerebro-cardiovascular disease, obstructive sleep apnea, hypogonadism, polycystic ovary syndrome, and various types of cancers [4,5]. Moreover, obesity is associated with many concomitant endocrine disorders of which their underlying mechanisms remain difficult to understand [6]. According to the literature, the loss of about 5% to 10% of body weight is associated with significantly improved obesity-related comorbidities; however, the treatment of obesity and its comorbidities constitutes a major challenge for clinicians [7].

Several strategies are available to drive weight loss, including modification of the lifestyle (both diet and physical activity), pharmacological intervention, and bariatric surgery [7]. Among the possible approaches available to achieve significant weight loss, the consumption of a very-low-calorie ketogenic diet (VLCKD) represents a promising option. A high-fat ketogenic diet has been used to treat refractory epilepsy in children since the 1920s [8,9]. In the last few years, VLCKD, an evolution of the first ketogenic diets, is increasingly being used in the clinical setting for the treatment of obesity. VLCKD generally is considered a second-line intervention in patients who do not respond to conventional dietary treatment [10]. This dietary intervention mimics fasting by ensuring a marked reduction in carbohydrates (i.e., the daily intake is usually lower than 30 g/day) [8], which leads to the synthesis of ketones [11]. These ketones are used as fuel by many tissues, including the central nervous system, skeletal muscle, and the heart [12]. Ketogenesis is the main mechanism behind the anorexigenic effect of VLCKD and helps to improve the compliance with and motivation to adhere to treatment [13]. Thus, severe obesity, characterized by a body mass index (BMI) of greater than 35 kg/m^2^, represents a major potential application for VLCKD [10]. 

The VLCKD weight-loss program is characterized by different phases. Usually, it includes a first step of calorie and carbohydrate restrictions, a second step of metabolic stabilization, and lastly, a maintenance phase [10]. VLCKD is part of a multidisciplinary approach to obesity and requires proper medical supervision [7]. Before starting VLCKD, patients must be carefully screened to reveal any contraindications to this treatment approach. Absolute contraindications include kidney failure, liver failure, heart failure (New York Heart Association functional classes III–IV), respiratory failure, unstable angina, recent stroke or myocardial infarction (<12 months), cardiac arrhythmias, eating disorders (e.g., anorexia nervosa, bulimia), mental illness, alcohol or substance abuse, and active and/or severe infections. In pregnant women or in those who are breastfeeding, VLCKD is also contraindicated. Moreover, patients with type 1 diabetes mellitus, β-cell failure in DM2, or who use sodium/glucose cotransporter 2 (SGLT2) inhibitors are also considered nonviable candidates for VLCKD due to the risk for euglycemic diabetic ketoacidosis in these populations [10]. 

VLCKD is increasingly being deployed in the clinical setting and a growing number of studies have suggested its effectiveness. A recent systematic review and meta-analysis including 12 studies and 801 patients reported significant efficacy of VLCKD in reducing body weight, BMI, and waist circumference [7]. However, even though several studies have investigated the efficacy of VLCKD in obese patients and the available data are encouraging, to our knowledge, no research team has yet investigated its effects on testicular hormonal function in overweight/obese male patients.

Therefore, the aim of the present study was to evaluate the effectiveness and the safety of VLCKD in overweight and obese male patients with regard to their metabolic and hormonal profiles. In particular, we focused our attention on elucidating the effects of this dietary intervention on the gonadal profile and on a biochemical marker of prostatic disease.

## 2. Materials and Methods 

### 2.1. Study Design

This prospective study enrolled 40 overweight/obese men (mean age: 45.8 ± 2.42 years) attending the Division of Andrology and Endocrinology, University Teaching Hospital “Policlinico-Vittorio Emanuele” University of Catania for weight-loss purposes. We excluded patients with a BMI of less than 26 kg/m^2^, kidney failure, liver failure, heart failure (New York Heart Association functional classes III–IV), respiratory failure, unstable angina, recent stroke or myocardial infarction (<12 months), cardiac arrhythmias, eating disorders (i.e., anorexia nervosa, bulimia), mental illness, alcohol or substance abuse, and active and/or severe infections. As mentioned previously, type 1 diabetes mellitus, β-cell failure in DM2, and the use of sodium/glucose cotransporter 2 (SGLT2) inhibitors were also considered exclusion criteria. 

Finally, before starting VLCKD, we excluded patients with hypercortisolism using a 1 mg overnight dexamethasone suppression test as a diagnostic assessment. 

### 2.2. Diet Protocol

The VLCKD protocol involved the marked restriction of daily carbohydrate intake (<30 g/day) and a proportional intake of fat (about 44%) and protein (about 43%). The daily protein intake in this context is 1.2 to 1.5 g/kg of the ideal body weight. Since this study was conducted involving male subjects, we chose a protein intake of around 1.4 to 1.5 g/day to support muscle mass.

The ketogenic diet plan is generally divided into five phases. In the first phase (VLCKD: 600–800 kcal/day), there is a total replacement of natural proteins with five protein preparations (breakfast, lunch, dinner, amid-morning snack, and a mid-afternoon snack) and the possibility of eating vegetables with low glycemic index values during lunch and dinner. Each replacement meal is made up of high biological value proteins (milk proteins, such as whey protein, at rapid absorption and caseins at low absorption; avian eggs; vegetable proteins such as soya, green peas, and cereals), lipids (monounsaturated fatty acids such as high grade sunflower oil; polyunsaturated fatty acids such as omega-3 and omega-6 fatty acids), carbohydrates (starch; sweeteners including aspartame, acesulfame, sucralose, and cyclamates; polyalcohols such as sorbitol, maltilolo, erythritol, and polydextrose), insoluble fibers (cellulose, hemy-cellulose, and lignin); and soluble fibers (galactooligosaccharides, fructooligosaccharides, and polysaccharides such as inulin). Animal proteins contain the totality of essential amino acids (phenylalanine, tryptophan, methionine, treonine, lysine, isoleucine, istidine, leucine, and valine). Replacement meals were purchased by patients from two specialized companies.

In the second phase (low-calorie ketogenic diet: 800–1000 kcal/day), two options can be chosen. The first is to replace a single protein preparation with a natural protein food (lunch or dinner), such as meat, eggs, or fish; the second option replaces both lunch and dinner with natural proteins and maintains breakfast and snacks as having the original protein preparations. In both cases, it is possible to eat vegetables only with low glycemic index values. During these two phases, which last about 12 weeks and which are the phases in which ketosis is maintained, integration with micronutrients is recommended. These consist of vitamins (complexes B, C, and E), minerals (sodium, potassium, magnesium, and calcium), and omega-3 fatty acids.

In the third phase (low-calorie diet: 1200–1500 kcal/day), carbohydrates are gradually replenished based on their increasing glycemic index values and at the same time, protein preparations are progressively replaced with natural foods, customizing the program according to the needs of the person. In addition, vegetables with greater carbohydrate content can be introduced. Meanwhile, the first foods to be reintroduced are fruit and dairy products. Breakfast and one snack are still made up of protein preparations, while the other snack is replaced by fruit and lunch and dinner are made up of natural proteins (meat, eggs, fish, and dairy products three to four times per week). During the reintroduction phase, the patient is required to conduct at least 10 min of physical activity per day and to progressively increase the amount of exercise performed according to their physical abilities.

In the fourth and fifth phases, pasta or bread (lunch), cereals (breakfast or dinner), and finally, legumes (lunch or dinner) are reinstated. During the last phase, there is a food plan that balances macro- and micro-nutrients with the total replacement of the protein preparation by natural foods and a daily intake of between 1500 and 2000 kcal/day, depending on the individual. At this point, 150 min/week of physical activity is always recommended since the main objective during the maintenance phase is the control of body weight.

In the present study, each patient adhered to VLCKD for at least eight weeks. Informed written consent was obtained from each study participant and the study was performed in accordance with the Declaration of Helsinki and was approved by the local ethics committee. Anthropometric parameters and biochemical data were evaluated before VLCKD was started and at the end of the second phase, when the patients were within the ketosis period. The state of ketosis was confirmed by using specific test strips for the measurement of ketonemia in capillary blood (Taidoc Technology Corporation, New Taipei City, Taiwan). All procedures described in this manuscript were part of common clinical practice and this study was approved by the intradivisional ethics committee of the Endocrinology Section.

### 2.3. Anthropometric Parameters and Biochemical Data

Body weight (kg) and height (m) were measured for each patient in a fasting state, without shoes and without wearing heavy clothes, by using the same calibrated scale and stadiometer. The waist circumference (cm) was also measured at the midpoint between the lower rib and iliac crest using a measuring tape with an accuracy of 0.1 cm.

The following biochemical data were collected both before and after VLCKD: fasting glucose, insulin, total cholesterol, high-density lipoprotein (HDL), triglycerides (TGL), creatinine, uric acid, aspartate aminotransferase (AST), alanine aminotransferase (ALT), vitamin D, luteinizing hormone (LH), total testosterone (TT), and prostate-specific antigen (PSA). We calculated the homeostasis model assessment (HOMA) index for the evaluation of insulin resistance and the low-density lipoprotein (LDL) level to complete the framework for the glycolipid profile. 

During the period of VLCKD, all patients underwent the collection of blood samples for discerning the control of electrolytes (e.g., sodium, potassium, and chlorine).

### 2.4. Statistical Analysis 

The results are reported as mean ± standard error of the mean throughout the study. Statistical analysis of the data was performed using the paired Student’s *t*-test. Differences in the percentage of patients with biochemical markers within the normal range before and after VLCKD were evaluated by the chi-squared test. Multivariate regression analysis was performed for body weight reduction. The Statistical Package for the Social Sciences version 22.0 for Windows software program (IBM Corporation, Armonk, NY, USA) and the RealStatistics add-on for Microsoft Excel (Microsoft Corporation, Redmond, WA, USA) were used for statistical analyses. The results with *p*-values of less than 0.05 were considered statistically significant.

## 3. Results

Table 1 summarizes the characteristics of all the patients enrolled in this study, while the results of analyses stratified according to obese or overweight patients are reported in Table 2. After VLCKD (13.5 ± 0.83 weeks), all patients showed a reduction in body weight, with a significant decrease in both BMI and waist circumference from baseline (*p* < 0.01). The average amount of weight loss was 21.05 ± 1.44 kg, exhibiting an 18% decrease relative to the baseline values (Table 1). 

At baseline, three patients (7.5%) had diabetes mellitus, 11 patients (27.5%) showed impaired fasting blood glucose (IFG), and 32 patients (80%) demonstrated insulin resistance (HOMA-index >2.5). After VLCKD, fasting glucose, insulin, and HOMA index values were significantly improved (*p* < 0.01) (Table 1). Interestingly, only one patient had a fasting glucose of 128 mg/dL (baseline value: 170 mg/dL), while all others (97.5%) presented values of less than 100 mg/dL. Moreover, 39 of 40 patients demonstrated normal HOMA index values after VLCKD (Table 3). 

Regarding the lipid profile and transaminases, total cholesterol, HDL, TGL, LDL, AST, and ALT were improved significantly after VLCKD (*p* < 0.01; *p* < 0.05 only for AST) (Table 1). In detail, at baseline, 20 patients (50%) showed total cholesterol values of greater than 200 mg/dL, eight patients (20%) had TGL levels of more than 200 mg/dL, and 17 patients had HDL levels of less than 40 mg/dL. After VLCKD, only five patients (12.5%) showed high total cholesterol levels and four (10%) had HDL levels of less than 40 mg/dL, while all enrolled patients presented TGL levels within the normal range.

Serum creatinine levels were increased slightly, but all patients maintained levels within the normal range with an average value of 0.84 ± 0.02 mg/dL, whereas uric acid levels were decreased, but not significantly, after VLCKD (Table 1).

Vitamin D, LH, and TT increased significantly (*p* <0.01), while the PSA significantly decreased as compared to baseline (*p* <0.01) (Figure 1). Prior to the dietary intervention, five patients (12.5%) had overt hypogonadism (TT <230 ng/dL); after VLCKD, four of them had fully recovered their hypothalamic–pituitary–testicular axis function.

Regarding adverse effects, one patient complained of urolithiasis after VLCKD. Further, 50% of participants experienced minor adverse effects in the first phase after starting VLCKD, but all these symptoms were mild and resolved spontaneously within a few days (Table 4). Patients showed good compliance with VLCKD and none of them dropped out of the study. Patient purchase of replacement meals probably played an important role in motivating them. Finally, serum electrolytes remained within the normal range.

Since the length of treatment was different for patients enrolled, we constructed a multivariate regression model, including the length of treatment (number of weeks during which patients were exposed to VLCKD), age, weight, BMI, glycemia, insulin, and HOMA index at baseline. Using a stepwise procedure, the number of weeks adhering to VLCKD was the only variable to significantly influence the percentage of body weight reduction. This parameter alone explains 18% of the outcome variability (Table 5).

## 4. Discussion

Obesity and obesity-related diseases including diabetes, cardio- and cerebrovascular disease, arterial hypertension, dyslipidemia, and cancer constitute a worrying health concern worldwide. Thus, interventions aimed at changing one’s lifestyle (e.g., including a shorter daily mobile telephone use could contribute to the reduction of body weight and facilitate the maintenance of a better testicular function [14]), diet protocols, drugs, and bariatric surgery have been proposed as useful strategies for weight loss. 

A high-fat ketogenic diet has been adopted since 1921, when it was used to treat epilepsy and migraines [8,9]. More recently, VLCKD emerged as a valid strategy available in the treatment of obesity. VLCKD is a dietary intervention characterized by a restriction in daily carbohydrate intake (<30 g/day) and a relative increase in the proportions of fat and protein. In the first stage, the total energy intake is lower than 800 kcal/day [6]. However, VLCKD is not a high-protein diet, since the daily protein intake is maintained around 1.2 to 1.5 g/kg of the ideal body weight [10]. Thus, VLCKD mimics fasting and leads to the onset of a state of ketosis. Ketone bodies act as anorexigenic agents, triggering a reduction in hunger and food intake [10]. However, the Italian Society of Endocrinology recommended VLCKD as a second-line option in the management of weight loss and for a maximum of 12 weeks [10].

To our knowledge, this is the first study to evaluate the effects of VLCKD on metabolic and LH/TT hormonal profiles in a population of overweight/obese male patients. After VLCKD, we found that all patients experienced a relevant decrease in their body weight, waist circumference, and BMI as compared with baseline. 

Notably, these results are in line with previous studies. A 2013 meta-analysis of 13 studies showed that body weight decreased significantly after VLCKD [15]. Similarly, Merra et al. studied the effects of three types of VLCKD (all with carbohydrates intake <50 g/day) in 54 overweight/obese patients, observing a significant reduction in BMI, waist circumference, and total body fat after three weeks of VLCKD [16]. Even more recently, a systematic review and meta-analysis of 12 studies, including a total of 801 patients, highlighted significant improvements in body weight, BMI, and waist circumference after VLCKD [7]. Bruci et al. have also recently reported similar results upon evaluating 92 consecutively obese patients undergoing VLCKD; among these individuals, after dietary intervention, body weight, BMI, and fat mass were significantly lower than at baseline [17].

Regarding the glycemic profile, we determined that all parameters that were assessed were improved significantly after VLCKD. Understanding the beneficial effects of this diet on glycemic homeostasis has been the goal of many studies. Bueno et al., in their meta-analysis, did not find statistically significant results [8]. On the contrary, in another very recent meta-analysis, Castellana et al. reported a significant decrease in glycosylated hemoglobin (HbA1c) after VLCKD, but the authors did not evaluate other parameters such as fasting glucose and insulin serum levels [7]. A recent systematic review and consensus statement from the Italian Society of Endocrinology has highlighted that VLCKD is an effective tool by which to obtain good glycemic control, reducing the fasting glucose, HbA1c, plasma insulin, and C-peptide levels and consequently, lowering the HOMA index [10]. This dietary intervention seems to improve ß-cell function; thus, the remission of diabetes is possible [10,18]. Accordingly, we found that after VLCKD, only one patient retained a fasting glucose plasma level of greater than 126 mg/dL and only one patient showed persistent insulin resistance. 

With regard to the lipid profile, we found that total cholesterol, LDL, and HDL levels were improved significantly after VLCKD. Moreover, we also observed a statistically significant decrease in TGL level and after the dietary intervention, all patients presented TGL values within the normal range. Data available on this topic are controversial, likely as a result of the differences in the diet composition prescribed. Volek and Sharma found a transient increase in total cholesterol and LDL levels after four weeks of VLCKD, while at the end of the observation period (eight weeks), both values were within their normal ranges and did not differ significantly from baseline. These authors also reported a nonsignificant trend for increased HDL and a marked decrease in TGL [19]. Similarly, in 2013, Bueno et al. observed a significant reduction in TGL and an increase in the HDL level, also highlighting a significant increase in LDL values [8]. More recently, Caprio et al. in their systematic review showed a beneficial effect of VLCKD on total cholesterol, LDL, HDL, and TGL values [10], while Castellana et al. recorded significant decreases in total cholesterol and TGL, but no changes in the LDL or HDL level [7]. 

In the present study, at the end of the VLCKD period, we did not observe any differences from baseline for the creatinine and uric acid levels. Four patients experienced mild, transient, and spontaneously resolving hyperuricemia at the initiation of the diet. However, transient hyperuricemia is a frequent side effect of the ketogenic diet [20]. Moreover, this dietary regimen is contraindicated in the case of kidney failure and moderate-to-severe kidney disease and all patients enrolled in our study were screened for renal function before starting VLCKD. However, a recent study suggests that under clinician supervision, this diet is an effective and safe tool against obesity that can also be used in patients with mild kidney failure (estimated glomerular filtration rate: 60–80 mL/min) [17].

A very interesting aspect to consider is the relationship between VLCKD and vitamin D. It is known that vitamin D levels are lower in obese people than in normal-weight subjects, possibly given volumetric dilution effects, but also due to other obesity-related factors, such as poor dietary intake, lack of exposure to sunlight, and lower skin synthesis [1,15]. Although the 2020 European Society of Endocrinology (ESE) guidelines about the work-up in obesity do not suggest the routine measurement of vitamin D levels in obese patients, its evaluation in patients undergoing VLCKD is of great interest since suboptimal levels seem to be associated with impaired glucose homeostasis, insulin resistance, and DM2 [15]. In our experience, we found that vitamin D levels become significantly increased after VLCKD. Perticone et al., in their randomized study of 28 patients allocated to VLCKD and 22 patients allocated to standard hypocaloric Mediterranean diets, found that vitamin D levels increased significantly only in the former group.

Obesity and low vitamin D levels (and other comorbidities that are obesity-related, such as metabolic syndrome and DM2) are also associated with a functional “dysmetabolic” hypogonadism [1,15,21]. In men with obesity, the prevalence of hypogonadism ranges between 22.9% and 78.8% [6]. However, according to the ESE guidelines, gonadotropin, TT, and sexual hormone binding globulin (SHBG) for the determination of bioavailable testosterone must be assessed only in the presence of known signs/symptoms of hypogonadism [1]. Obesity was suggested to be associated with low or inappropriately normal levels of LH, indicating the occurrence of a dominant suppression at the hypothalamic–pituitary level [22]. To our knowledge, no studies have yet investigated the effects of VLCKD on testicular hormonal function in overweight/obese male patients. We herein evaluated LH, TT, and PSA levels before and after VLCKD and at the end of the dietary intervention, we found that the LH and TT levels were increased significantly, while only one subject had hypogonadism. These results suggest that VLCKD may unblock the hypothalamic–pituitary–gonadal axis, stimulating the secretion of LH. LH binds to receptors on Leydig cells, stimulating T secretion. In addition, it has been demonstrated that the testes express most of the enzymes involved in vitamin D activation [23]. Thus, the increased levels of LH also stimulate the 25-hydroxylation activity of Leydig cells, with a consequent increase in vitamin D levels. These findings are very interesting as if obesity can lead to hypogonadism, it is also true that testosterone deficiency is associated with impaired glucose homeostasis, insulin resistance, metabolic syndrome, and DM2 [24]. In the present study, we also assessed for the first time total PSA levels and we found that a statistically significant decrease occurred after VLCKD. Notably, this is an interesting and promising result. In the past several years, different studies have shown that impaired glucose homeostasis, hyperinsulinemia, and insulin resistance increase the risks of both benign prostatic hyperplasia and severe prostate inflammation [25]. According to this evidence, we conducted a cross-sectional study of 544 consecutive patients with benign prostatic hyperplasia and related low urinary tract symptoms (LUTS) and found that patients with hyperinsulinemia and insulin resistance had a greater risk of experiencing severe LUTS and erectile dysfunction. Thus, metabolic syndrome and hyperinsulinemia should be considered as prospective targets of therapy to counteract the associated prostate overgrowth [26]. However, further studies in this regard are needed.

Finally, regarding the adverse effects of VLCKD, we found that half of enrolled patients complained of mild and transient side effects, mainly consisting of gastrointestinal symptoms (constipation, diarrhea, pyrosis), asthenia, headache, and hyperuricemia. A single patient had long-term side effects (urolithiasis). Our findings are in agreement with other data available in the literature. According to a 2019 practical guide, the most frequent short term side effects of VLCKD include dehydration, hypoglycemia, lethargy, halitosis, nausea/vomiting, constipation, diarrhea, gastroesophageal reflux disease, and hyperuricemia [20]. Meanwhile, long term side effects encountered may include hypoproteinemia, hypocalcemia, bone damage, LDL increase, urolithiasis, gallstone, and hair loss [20]. Nevertheless, under strict medical supervision, VLCKD may be considered safe both in the short and long term [7,10]. 

## 5. Conclusions

Obesity constitutes a clinical condition that has a series of consequences on lifestyle habits, which in turn amplify the negative impact of obesity itself on general health. Because of its faster control of body weight compared with the traditional diet, VLCKD exerts an extraordinary motivational drive that is capable of removing other acquired risk factors that are obesity-related, such as a sedentary lifestyle or depression. Reduced testicular function is often found in obese patients; given its systemic implications, this could represent an additional element that is capable of feeding this vicious circle. The results of the present study suggest that VLCKD is an effective tool against obesity. We observed significant improvements in glucose homeostasis and the lipid profile. Moreover, significant increases in vitamin D, LH, TT, and PSA levels were observed, suggesting that VLCKD could be a noninvasive and effective therapeutic strategy in dysmetabolic patients with hypogonadism and LUTS.

## 6. Limitations of the Study 

This study was hampered by its small sample size, lack of a control group on a nonketogenic low calorie diet, lack of sex hormone binding globulin (SHBG) measurements that could have helped to better assess the testicular function, and lack of prostate ultrasound by which to evaluate any changes in the prostate gland parenchyma or reduction of the prostatic volume. However, any comparison with the Mediterranean diet has little scientific value as the comparison between a preventive diet model (the Mediterranean one) and a diet therapy limited in time and aimed at rapid weight loss has no rationality. The Mediterranean diet, while inducing general health benefits, does not seem to be superior to other types of interventions if used to produce weight loss in overweight or obese patients.

Therefore, further and larger studies are needed to better evaluate the effectiveness of VLCKD on testicular function and prostate volume.

## Figures and Tables

**Figure 1 nutrients-12-02967-f001:**
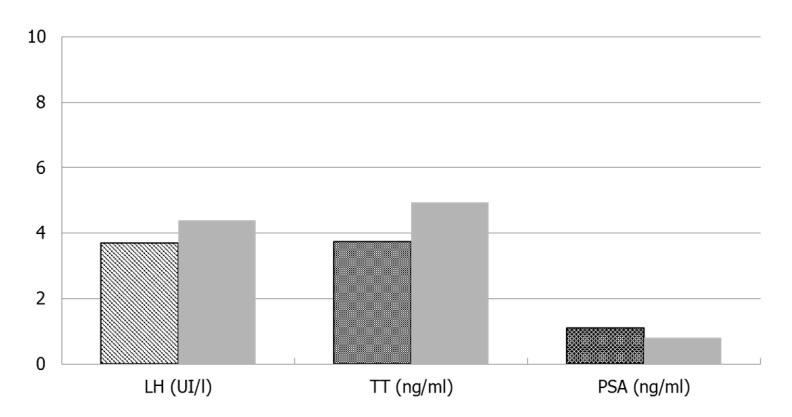
Luteinizing-hormone (LH), total testosterone (TT), and prostatic-specific antigen (PSA) before and after VLCKD.

**Table 1 nutrients-12-02967-t001:** Anthropometric, metabolic, and hormonal parameters before and after very low calorie ketogenic diet (VLCKD).

Parameter	Before VLCKD	After VLCKD	Normal Values
Weight (kg)	112 ± 3.7	90.9 ± 2.9 *	NA
WC (cm)	126.8 ± 2.2	104.2 ± 2.1 *	<102
BMI (kg/m^2^)	37.5 ± 1.1	30.5 ± 0.9 *	19.5–24.9
Glucose (mg/dL)	96.1 ± 3	84.6 ± 1.8 *	60–100
Insulin (µUI/mL)	20.5 ± 2.1	6.1 ± 0.4 *	1.9–23
HOMA index	4.9 ± 0.6	1.3 ± 0.1 *	0.23–2.5
Total cholesterol (mg/dL)	204.3 ± 8.2	166.3 ± 4.6 *	<200
HDL (mg/dL)	42 ± 1.4	48.8 ± 1.7 *	>48
LDL (mg/dL)	130.9 ± 6.8	100.9 ± 4.3 *	NA
Triglycerides (mg/dL)	156.8 ± 16.4	83.2 ± 4.4 *	<150
AST (U/L)	34.1 ± 4.8	23.4 ± 1.2 ^†^	<35
ALT (U/L)	39.4 ± 4.8	25.8 ± 1.8 *	<35
Creatinine (mg/dL)	0.82 ± 0.02	0.84 ± 0.02	0.51–0.95
Uric acid (mg/dL)	5.7 ± 0.2	5.5 ± 0.2	2.4–5.7
Vitamin D (mg/L)	19.9 ± 1.1	38.5 ± 1.8 *	30–100

* *p* < 0.01; ^†^
*p* < 0.05, Legend: NA = not applicable; WC = waist circumference; BMI = body mass index; HOMA = Homeostasis Model Assessment; HDL = high-density lipoprotein; LDL = low-density lipoprotein; AST = aspartate aminotransferase; ALT = alanine aminotransferase

**Table 2 nutrients-12-02967-t002:** Anthropometric, metabolic, and hormonal parameters before and after very low calorie ketogenic diet (VLCKD) in obese and overweight patients.

	Obese (*n* = 34)		Overweight (*n* = 4)	
Parameter	Before VLCKD	After VLCKD	Before VLCKD	After VLCKD
Weight (kg)	115.2 ± 3.8	93.0 ± 3.1 ^†^	85.2 ± 3.8	73.3 ± 3.2 *
WC (cm)	128.2 ± 2.4	105.5 ± 2.3 ^†^	114.3 ± 1.4	93.0 ± 1.8 ^†^
BMI (kg/m^2^)	38.6 ± 1.1	31.2 ± 0.9 ^†^	28.4 ± 0.4	24.5 ± 0.6 *
Glucose (mg/dL)	97.0 ± 3.3	85.3 ± 2.0 ^†^	87.8 ± 2.3	78.8 ± 2.3 *
Insulin (µUI/mL)	21.3 ± 2.2	6.1 ± 0.4 ^†^	13.7 ± 4.2	6.1 ± 1.5
HOMA index	5.2 ± 0.6	1.3 ± 0.1 ^†^	2.9 ± 0.9	1.2 ± 0.3
Total cholesterol (mg/dL)	203.4 ± 9.1	166.8 ± 5.0 ^†^	211.3 ± 8.5	162.8 ± 10.7 *
HDL (mg/dL)	42.2 ± 1.5	48.8 ± 1.8 ^†^	40.5 ± 2.6	49.0 ± 5.0
LDL (mg/dL)	129.4 ± 7.5	101.5 ± 4.7 ^†^	143.4 ± 8.8	96.1 ± 10.3 *
Triglycerides (mg/dL)	159.1 ± 18.2	83.5 ± 4.6 ^†^	137.0 ± 22.8	88.5 ± 16.6 *
AST (U/L)	35.7 ± 5.3	24.0 ± 1.2 *	20.5 ± 3.4	18.3 ± 3.1 *
ALT (U/L)	41.1 ± 5.3	25.9 ± 1.9 ^†^	24.8 ± 5.9	21.0 ± 4.1
Creatinine (mg/dL)	0.80 ± 0.02	0.84 ± 0.02*	0.89 ± 0.00	0.93 ± 0.03
Uric acid (mg/dL)	5.7 ± 0.2	5.6 ± 0.2	5.7 ± 1.0	5.0 ± 0.8
Vitamin D (mg/L)	20.6 ± 1.1	39.2 ± 1.9 ^†^	14.3 ± 1.9	32.5 ± 2.9 ^†^
LH (IU/L)	3.7 ± 0.3	4.3 ± 0.2 ^†^	3.6 ± 0.2	4.5 ± 0.5
TT (ng/dL)	382.0 ± 21.2	498.4 ± 30.4 ^†^	357.0 ± 27.8	471.3 ± 38.4 ^†^
PSA (ng/mL)	1.1 ± 0.1	0.9 ± 0.1 ^†^	1.0 ± 0.2	0.5 ± 0.1

* *p* < 0.01 vs. before VLCKD; ^†^
*p* < 0.05 vs. before VLCKD. Legend: WC = waist circumference; BMI = body mass index; HOMA = Homeostasis Model Assessment; HDL = High-density lipoprotein; LDL = Low-density lipoprotein; AST = aspartate aminotransferase; ALT = alanine aminotransferase; LH = luteinizing hormone; TT = total testosterone; PSA = prostate-specific antigen.

**Table 3 nutrients-12-02967-t003:** Percentage of patients with biomarkers within the normal range.

Parameter	Before VLCKD	After VLCKD
WC < 102 cm	0% (0/38)	55.3% (21/38)
BMI < 29.9 kg/m^2^	13.2% (5/38)	55.3% (21/38) *
BMI < 24.9 kg/m^2^	0% (0/38)	13.2% (5/38)
Glucose < 100 mg/dL	57.9% (22/38)	97.4% (37/38) *
Insulin < 23 µUI/mL	63.2% (24/38)	100.0% (38/38)
HOMA index < 2.5	15.8% (6/38)	97.4% (37/38) *
Total cholesterol < 200 mg/dL	42.1% (16/38)	86.8% (33/38) *
HDL > 48 mg/dL	23.7% (9/38)	36.8% (14/38)
Triglycerides < 150 mg/dL	57.9% (22/38)	97.4% (37/38) *
AST < 35 U/L	65.8% (25/38)	91.9% (34/37) *
ALT < 35 U/L	52.6% (20/38)	83.8% (31/37) *
Creatinine < 0. 95 mg/dL	86.5% (32/37)	73.0% (27/37)
Uric acid < 5.7 mg/dL	56.8% (21/37)	59.5% (22/37)
Vitamin D > 30 mg/L	5.4% (2/37)	81.1% (30/37) *
LH (IU/L) < 8.75 IU/L	100.0% (38/38)	100.0% (38/38)
TT > 230 ng/dL	86.8% (33/38)	97.4% (37/38)
PSA < 4 ng/mL	100.0% (38/38)	100.0% (38/38)

* *p* < 0.05 vs. before VLCKD, chi-squared test. Legend: WC = waist circumference; BMI = body mass index; HOMA = Homeostasis Model Assessment; HDL = high-density lipoprotein; AST = aspartate aminotransferase; ALT = alanine aminotransferase; LH = luteinizing hormone; TT = total testosterone; PSA = prostate-specific antigen.

**Table 4 nutrients-12-02967-t004:** Regression analysis.

	Coeff	St. Error	T Stat	*p*-Value	Lower	Upper
Intercept	11.821	2.49438	4.73905	<0.01	6.76217	16.8798
Weeks	0.49163	0.17294	2.84274	<0.01	0.14089	0.84238

Legend: Coeff = Coefficient; St. Error = Standard error.

**Table 5 nutrients-12-02967-t005:** Adverse effects during the consumption of a very low calorie ketogenic diet.

Sign/Symptom	No. of Patients	Percentage
Headache	5	12.5%
Asthenia	3	7.5%
Constipation/diarrhea	4	10%
Joint pain	1	2.5%
Urolithiasis	1	2.5%
Pyrosis	3	7.5%

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
