# Peer review of "Effectiveness of a Very Low Calorie Ketogenic Diet on Testicular Function in Overweight/Obese Men"

_nutrients, 2020, doi:10.3390/nu12102967_

Round 1

Reviewer 1 Report

This study examines the effectiveness of VLCKD among 40 obese/ overweight men who received at least 8 weeks treatment.  I am worried about your statistical method. In addition, A number of limitations are noted that reduce the ability to discern the impact of this study on the field:

Intro:

Intro is not focused and should be reconstructed. More information is needed to describe VLCKD. Growing number of studies including RCTs and a few systematic reviews has shown the effectiveness of VLCKD. I would suggest paying more attention to those studies including findings and weakness to increase the significance of the current study.

Line 40-42 and paragraph are confusing.

Method:

If have a lot of exclusion criteria, please provide some rationale. Again, this could be part of literature review in the intro.

More information is needed for the t test in the statistical analysis section. I am unable to tell if you used independent or paired t-tests. You examined the change of biomarkers for the same subjects before and after the study. A paired t-test should be used when we are interested in the difference between two variables for the same subject. I would suggest two more sensitivity analyses. One is stratified analyses among obese patients and among overweight. Additionally, given these enrolled sample receiving different length of treatment, a regression adjusted for length of treatment may increasing the robustness of this study.

Results:

I would suggest providing one additional table for percentage of patients whose biomarkers are within normal range. For example, you reported “At baseline, 3 patients (7.5%) had diabetes mellitus, 11 patients (27.5%) showed impaired fasting blood glucose (IFG) and 32 men (80%) had insulin-resistance (HOMA-index >2.5).” but readers cannot find the result from the table.

Discussion:

As your discussion is all about comparison with previous study, more justification of potential reasons of the finding is needed.

A weakness section is need (e.g. single-arm uncontrolled trial, small sample size, different length of treatment, etc.).

Author Response

Comment 1:  Intro is focused and should be reconstructed. More information is needed to describe VLKCD. Growing number of studies including RCTs and a few systematic reviews has shown the effectiveness of VLKCD. I would suggest paying more attention to those studies including findings and weakness to increase the significant of the current study.

Answer to comment 1: As suggested, the Introduction was reconstructed. The basis of VLKCD are now also described in the revised Introduction. We underline that, even if several studies investigated the efficacy of VLCKD in obese patients and data are encouraging, to our knowledge, no studies have investigated the effects of VLCKD on testicular hormonal function in overweight/obese male patients. The details of the various studies cited are argued in the Discussion.

Comment 2: Line 40-42 and paragraph are confusing.

Answer to comment 2: We modified this paragraph. Please see lines 40-42. 

Comments 3:  I have a lot of exclusion criteria, please provide some rationale. Again, this could be part of literature review in the intro.

Answer to comment 3: The exclusion criteria are related to the contraindication of VLKCD that we added in the revised version of the introduction (please see lines 64-70). 

Comment 4: More information is needed for the t test in the statistical analysis section. I am unable to tell if you used independent or paired t-test. You examined the change of biomarkers for the same subjects before and after the study. A paired t-test should be used when we are interested in the difference between two variables for the same subject.

Answer to comment 4: The paired t-test was used. This has now been specified in the methods section.

Comment 5: I would suggest two more sensitivity analyses. One is stratified analyses among obese patients and among overweight.

Answer to comment 5: Thank you for this comment. Although the number of overweight patients is only 4 (which limits the reliability of the analysis), the results have been reported in Table 2. 

Comment 6: Additionally, given these enrolled sample receiving different length of treatment, a regression adjusted for length of treatment may increase the robustness of this study.

Answer to comment 6: A multivariate regression model including the length of treatment, age, weight, BMI, glycaemia, insulin, and HOMA-index values at baseline was build. Using a stepwise procedure, we were able to establish that the number of weeks was the only variable to significantly influence the percentage in body reduction after VLCKD, alone explaining the 18% of the body weight reduction.

Comment 7: I would suggest providing one additional table for percentage of patients whose biomarkers are within the normal range.

Answer to comment 7: Percentage of patients with biomarkers within the normal range, before and after VLCKD, has been added. This is the Table 3 of the revised version of the manuscript. Chi-square analysis was used to assess significant differences (this has been added in Methods).

Comment 8: As your discussion is all about comparison with previous study, more justification of potential reasons of the finding is needed.

Answer to comment 8: We better discussed the biological mechanisms that support our results (please see lines 306-315)

Comment 9: A weakness section is need (e.g. single-arm uncontrolled trial, small sample size, different length of treatment, etc.).

Answer to comment 9: We added a limitation paragraph (please see lines 350-359).

Reviewer 2 Report

This research is very interesting and helping to address many issues of concern regarding the Ketogenic diet; and its potential as one of the solution to the public to control body weight and diabetes. The study deign is very well conducted with only one concern regarding the sample size. However, I still believe that there is a need to expand the introduction by adding more literature review so that the history and understanding of ketogenic diet presented to the reader from beginning.

Author Response

Comments 1: This research is very interesting and helping to address many issues of concern regarding the ketogenic diet; and its potential as one of the solution to the public to control body weight and diabetes.

Answer to comment 1: We thank the Reviewer for her/his appreciation of this article.

Comments 2: The study design is very well conducted with only one concern regarding the sample size.

Answer to comment 2: As we wrote, the sample size is a limitation of our study (please see lines 350-359). The main reason for this relates to the lower number of men that accept to undergo VLCKD compared to women. In addition, some did not accept to blood withdrawn for andrological check-up. Nevertheless, to our knowledge, this is the first study that investigated the effects of VLCKD on the gonadal profile on obese/overweight men.

Comments 3: However, I still believe that there is a need to expand introduction by adding more literature review so that the history and understanding ketogenic diet presented to the reader from beginning.

Answer to comment 3: As suggested, the Introduction was reconstructed.

Round 2

Reviewer 1 Report

Authors have approximately addressed mu comments.